# Few-Shot Data-Driven Algorithms
# for Low Rank Approximation

**Piotr Indyk**
MIT
indyk@mit.edu

**Tal Wagner**
Microsoft Research Redmond
tal.wagner@gmail.com

**David P. Woodruff**
Carnegie Mellon University
dwoodruf@cs.cmu.edu

## Abstract

Recently, data-driven and learning-based algorithms for low rank matrix approximation were shown to outperform classical data-oblivious algorithms by wide margins in terms of accuracy. Those algorithms are based on the optimization of sparse sketching matrices, which lead to large savings in time and memory during testing. However, they require long training times on a large amount of existing data, and rely on access to specialized hardware and software.

In this work, we develop new data-driven low rank approximation algorithms with better computational efficiency in the training phase, alleviating these drawbacks. Furthermore, our methods are interpretable: while previous algorithms choose the sketching matrix either at random or by black-box learning, we show that it can be set (or initialized) to clearly interpretable values extracted from the dataset.

Our experiments show that our algorithms, either by themselves or in combination with previous methods, achieve significant empirical advantages over previous work, improving training times by up to an order of magnitude toward achieving the same target accuracy.

## 1 introduction

Low-rank approximation (LRA) of matrices is a fundamental operation in data analysis and machine learning, forming the basis for many efficient algorithms that involve storing and analyzing large matrices. Its importance has led to a large body of work on algorithms for computing LRA. While it is well-known how to find an optimal LRA of a given matrix in polynomial time using the Singular Value Decomposition (SVD), the degree of that polynomial makes the algorithm too slow for large matrices, defeating the purpose of leveraging it to design fast and scalable algorithms. With this motivation in mind, a long and prominent line of research has discovered *fast approximate* LRA algorithms, laying the foundations for a rich and expansive theory of fast algorithms for numerical linear algebra (e.g., [26, 51, 15, 16, 18, 24]). Due to their strong practical performance, these algorithms have become standard and widely used tools in machine learning over massive datasets (e.g., [57, 30, 43, 27, 13, 53]). See the surveys [41, 56, 42] for a comprehensive overview of this area.

These algorithms are mostly based on *sketching*. In its most basic form, the input matrix $A \in \mathbb{R}^{n \times d}$ is left-multiplied by a carefully chosen *sketching matrix* $S \in \mathbb{R}^{m \times n}$, whose row dimension is much smaller than that of $A$ (i.e., $m \ll n$). Then, a good low rank approximation to $A$ can be computed from $SA$. Reducing the row dimension from $n$ to $m$ improves the algorithmic efficiency in multiple ways. First, $SA$ can be stored using less space than $A$, which is important in the context of space-efficient streaming algorithms. Furthermore, the dimensionality reduction step can be combined with other steps to obtain LRA algorithms with running time nearly linear in the size of $A$. Identifying suitable sketching matrices is a highly non-trivial task, and by now there are several

known randomized constructions, including sparse sign matrices [15, 16, 43, 46], random projection matrices [51, 2], and subsampled Hadamard matrices [21, 13].

Unfortunately, a notable downside of sketching-based algorithms is that they lead to a larger approximation error than the optimal LRA, due to the lossy nature of sketching. Since this error propagates downstream to all tasks that use LRA as a subroutine, minimizing this error is highly important. This raises the question — can we design better sketching matrices, that offer a better tradeoff between the sketch width $m$ and the approximation quality? Such an improvement would directly translate into a more accurate (or alternatively, more space efficient) streaming algorithm for LRA (see Section 1.2 for more details). Furthermore, it would also be an important step towards obtaining more time-efficient algorithms as well.

**Data-driven approach to LRA.** A natural approach to the above challenge is to rely on the fact that in many cases, we have access to large quantities of past data (seen as a training set), which is related to the future input matrices (seen as a test set). In such a scenario, we could potentially use a preprocessing (or learning) stage on the past data in order to better handle future data. This approach, often called *data-driven* or *learning-augmented* algorithm design, has gained increased popularity lately, and led to improved algorithms for a host of classical problems (see Section 1.3).

For low rank approximation, this line of work was recently pioneered by [33], who proposed a data-driven LRA algorithm that we refer to henceforth as **IVY**. Specifically, IVY learns the values of the non-zero entries in a sparse sketching matrix $S$, instead of choosing them at random (as in classical LRA, e.g., [16]). Learning is done using stochastic gradient descent (SGD), by minimizing the empirical (normed) difference between matrices from the training set and their corresponding LRAs computed using $S$. Computing gradients for this loss function is a highly non-trivial operation, but nonetheless, [33] show it can be made tractable by exploiting modern auto-gradient software and GPU hardware, which are typically used nowadays to train deep neural networks.

**Challenges and limitations.** Despite its improved accuracy, the IVY algorithm has several drawbacks related to its computational cost. First, it requires a differentiable implementation of the SVD, which is slower than standard implementations, leading to long training times. Second, it relies on specialized software and hardware (automatic gradient computation in PyTorch, and access to a GPU). Third, in order to gain significant improvement over the oblivious (non-data-driven) baseline, it requires training over dozens of existing matrices, which may not necessarily be available. Fourth, it chooses the sketching matrix in a black-box, non-interpretable way, using black-box learning. All these limitations are also shared by [3, 38]. This leads us to consider whether much more efficient data-driven algorithms can attain a comparable advantage in their LRA accuracy.

## 1.1 Our Results

In this work, we propose a new approach to data-driven LRA, motivated by a different and more analytic-minded viewpoint. As mentioned above, previous work takes a "black-box" approach to the problem, directly minimizing the empirical loss via SGD. However, since LRA is a well-studied topic with a rich underlying mathematical structure, it is worth to consider alternative approaches that make direct use of our analytic understanding. This may allow us to simplify the learning algorithm — for example, by devising more effective loss functions — and perhaps even eliminate the heavy learning machinery altogether, in favor of analytic techniques.

This approach leads us to data-driven algorithms that are considerably faster to train than IVY, do not necessarily require access to specialized hardware and software, and use only one or few training matrices (thus, they perform *few-shot learning*). Specifically, we give the following two algorithms:

**One-shot closed-form algorithm:** Our first algorithm does not require learning it all; it only picks a random matrix from the dataset, and extracts a sketching matrix by an analytic computation. It requires neither GPU access nor autograd functionality.

**Few-shot SGD algorithm:** For our second algorithm, we use a new loss function for learned LRA (given in eq. (3) in Section 2). It is faster to compute and differentiate than the empirical loss used in [33], being a low degree polynomial, and can be (locally) optimized with only 2 or 3 steps of SGD, requiring only that many randomly chosen matrices from the dataset.

Both algorithms achieve significantly better sketch length to accuracy tradeoffs than data-oblivious LRA. When compared to IVY and its variants, they do attain similar (and eventually better) accuracy than our algorithms if allowed to run for sufficiently many SGD iterations, but with much longer training times and many more training matrices. In addition, our closed-form algorithm is also interpretable, as it computes a sparse sketching matrix $S$ with a clear mathematical meaning. Specifically, the non-zero entries of $S$ are set to be singular vectors of a block-partitioned randomly chosen training matrix, with a block partition determined by the sparsity pattern of $S$. The singular vectors in each block are chosen either greedily, or at random proportionally to their squared singular values.

Finally, we experiment with combining our approach with IVY, by initializing the weights of the sketching matrix according to our one-shot algorithm, instead of at random as in [33]. The result is an algorithm that uses the same memory, time and training samples as IVY, with significantly improved accuracy.

One takeaway message for designing data-driven algorithms, is that while it may seem tempting to invoke heavy-weight machine learning technology, there are possibly much simpler algorithms that attain similar performance with a significantly reduced computational cost, and an analytic understanding of the problem may help us arrive at those algorithms.

**Limitations of our approach.** Like most previous methods [33, 2], we focus on optimizing the non-zero entries of a sparse sketching matrix with a fixed sparsity pattern (chosen at random). Learning the sparsity pattern from data during training may improve performance even further. However, since the search space is discrete and very large, it is not clear how to do so without a very large increase in training time (more in section 3). We leave this as an important direction for future work.

**Broader impact.** Incorporating data into algorithms entails risks related to bias and fairness, as well as environmental effects of massive processing. Our methods carry benefits of interpretability and of significantly reduced GPU utilization, which may help alleviate these concerns. However, our work is focused on a generic algorithmic operation, and any application to a specific data domain would require further consideration of possible bias in the data as well as of privacy concerns.

## 1.2 Preliminaries

**Low rank matrix approximation.** In the LRA problem, the input is a matrix $A \in \mathbb{R}^{n \times d}$, say with $n \geq d$, and an integer $0 < k \ll d$. The goal is to output a matrix $A' \in \mathbb{R}^{n \times d}$ of rank $k$ which minimizes the approximation error $\|A - A'\|_F^2$, where $\|\cdot\|_F$ denotes the Frobenius matrix norm. It is a standard fact (known as the Eckhart-Young-Mirsky Theorem) that an optimal solution, often denoted as $A_k$, can be computed deterministically in time $O(nd^2)$ via the singular value decomposition. A long line of work has focused on algorithms that incur slightly larger approximation — namely $\|A - A_k\|_F^2 + \epsilon\|A\|_F^2$, or better yet $(1 + \epsilon)\|A - A_k\|_F^2$, for an arbitrarily small $\epsilon > 0$ — and run in time nearly linear in the size of $A$, and polynomial in $k$ and $\epsilon^{-1}$. See [56] for a survey.

**The SCW algorithm.** The data-oblivious LRA algorithm used as the basis for [33], henceforth called SCW, is due to [16], building on [51, 15]. It uses a random sketching matrix $S \in \mathbb{R}^{m \times n}$ (where $m \ll n$ is called the *sketching dimension*), chosen such that each column contains one non-zero entry at a uniformly random row, with a value which is uniformly random in $\{-1, 1\}$. Given an input matrix $A$ and the sketching matrix $S$, SCW computes the best LRA of $A$ in the rowspace of $A$. This can be computed in several equivalent ways, for example, by Algorithm 1 below.

---

**Algorithm 1:** Sketch-based matrix low-rank approximation

---

**Input:** Input matrix $A \in \mathbb{R}^{n \times d}$, sketching matrix $S \in \mathbb{R}^{m \times n}$.
**Output:** Rank-$k$ approximation $\tilde{A} \in \mathbb{R}^{n \times d}$ of $A$.

1 **Procedure** SCW($S, A$)*:*
2     $\tilde{U}\tilde{\Sigma}\tilde{V}^\top \leftarrow$ SVD of $SA$
3     $[A\tilde{V}]_k \leftarrow$ Optimal rank-$k$ approximation of $A\tilde{V}$ (computed by full SVD)
4     **return** $[A\tilde{V}]_k\tilde{V}^\top$

---

It is proven in [16] that a sketching dimension of $m = \text{poly}(k, \epsilon^{-1})$ suffices for the output rank-$k$ matrix $A'$ to satisfy $\|A - A'\|_F^2 \leq (1 + \epsilon)\|A - A_k\|_F^2$ with high probability. Furthermore, the algorithm can be implemented as a two-pass streaming algorithm, using space $O(m(n + d))$. Thus, reducing the sketch length $m$ directly reduces the storage requirements of the algorithm.

**Data-driven LRA.** The data-driven version of the LRA problem, introduced in [33], has two phases. In the training phase, we are given a training set of matrices, $\mathcal{A}_{\text{train}} \subset \mathbb{R}^{n \times d}$. Our goal is to construct, or learn, a sketching matrix $S$. In the testing phase, we are given a test set of matrices, $\mathcal{A}_{\text{test}} \subset \mathbb{R}^{n \times d}$. We compute a rank-$k$ approximation for each of them using the SCW algorithm (Algorithm 1), but with the *learned* sketching matrix from the training phase instead of a *random* sketching matrix. The goal is to choose $S$ so as to minimize the testing error,

$$\text{Err}(S, \mathcal{A}_{\text{test}}) := \sum_{A \in \mathcal{A}_{\text{test}}} \|A - \text{SCW}(S, A)\|_F^2. \tag{1}$$

**The IVY algorithm.** In [33], $S$ is initialized by sampling it from the same distribution as SCW – sparse random signs. The sparsity pattern (i.e., the locations of the non-zero entries in $S$) remains fixed, while the values of those entries are learned by SGD over the training loss,

$$S = \text{argmin}_{S'} \text{Err}(S', \mathcal{A}_{\text{test}}) = \text{argmin}_{S'} \sum_{A \in \mathcal{A}_{\text{test}}} \|A - \text{SCW}(S', A)\|_F^2. \tag{2}$$

This requires computing gradients over the steps of SCW, which in turn uses two invocations of the SVD (in steps 2 and 3 of Algorithm 1). While not all implementations of the SVD are differentiable, IVY employs a differentiable one which is based on the power method.

**Safeguard guarantees.** A common desideratum of data-driven algorithms is to guarantee they never perform much *worse* than their data-oblivious counterparts (see, e.g., [47, 44]), which could happen for example by over-tailoring to an irrelevant training set. For IVY, [33] note this can be achieved by simply concatenating an oblivious sketching matrix to the learned sketching matrix. This property is shared by subsequent work on data-driven LRA [3, 38] as well as by our algorithms.

## 1.3 Related Work

Most relevant to us are the recent papers [3, 38], who proposed modifications to IVY. Ailon et al. [3] study replacing dense linear layers in deep networks by *butterfly* gadgets, which are dense matrices with an implicit sparse decomposition (see Section 3 for details). For data-driven LRA, they propose replacing the random sparsity pattern used in IVY with the butterfly (implicit) sparsity pattern.

Liu et al. [38] suggest several extensions to [33]. First, they apply the IVY learning scheme to the LRA algorithm of [6], which has better asymptotic running time than SCW. Second, they suggest a greedy initialization method for $S$, instead of the random initialization used in IVY. Third, they apply the method to several other linear algebraic problems, for which they provide provable guarantees.

An LRA algorithm by [18] forms a data-dependent sketching matrix $S_A$ from $A$ by taking the top $k$ right-singular vectors of the product $SA$, where $S$ is an oblivious sketching matrix, and returns $AS_A S_A^\top$ as an LRA of $A$. Though not intended as such, it can be viewed as a data-driven LRA algorithm, if one forms $S_A$ from $A$ and then returns $BS_A S_A^\top$ as the LRA for a different input matrix $B$. However, this method cannot be safeguarded for worst-case guarantees as discussed above, and thus can perform arbitrarily poorly for $B$.

In [19], the random sketching matrices used by SCW are derandomized in a data-dependent fashion. While this leads to a data-dependent LRA algorithm, its goal is not to improve the error achieved by SCW, but rather to match it without using randomness. Indeed, its output sketching matrix is from the same set as SCW (sparse with $\pm 1$'s in its nonzero entries), while the data-driven LRA algorithms we study reduce the error by optimizing the non-zero values of the sketching matrix beyond $\pm 1$.

In a broader point of view, our work belongs to a recent trend of improving the performance of resource-constrained algorithms by taking a data-driven approach, which allows the algorithm to "prepare in advance" on a training set. In addition to LRA, other problems studied in this setting include caching [40, 50], scheduling [47, 37], data indexing [36, 25], set membership [44, 49, 54], frequency estimation over data streams [31, 1, 34, 17], clustering [8, 38], similarity search [20], support size estimation [22], multigrid methods [39], and more [29, 11, 9, 10, 5, 28, 35, 12, 55].

# 2 Few-Shot LRA Algorithms

In this section we present our proposed algorithms for few-shot data-driven LRA. The starting point of our algorithms is the same as IVY, in that they choose a sparse matrix $S_0 \in \mathbb{R}^{m \times n}$ where each column contains a single non-zero entry in a uniformly random row. The goal is to use the given training set to set the value of the non-zero entries.

## 2.1 One-Shot Closed-Form Algorithms

Our first approach involves no learning mechanisms — instead, it operates in a closed-form manner on a single randomly chosen matrix $A$ from the training set. We describe two algorithmic variants of this approach, leading to two of our algorithms, which we call **1Shot1Vec** and **1Shot2Vec**.

The sparsity pattern of $S_0$ partitions the rows of $A$ into blocks, $A^{(1)}, \ldots, A^{(m)}$, as follows: Let $I_i = \{j : S(i,j) = 1\}$, i.e., $I_i$ contains the non-zero column indices in row $i$ of $S$. The block $A^{(i)} \in \mathbb{R}^{|I_i| \times d}$ is the sub-matrix of $A$ that contains the rows whose indices are in $I_i$. Observe that in the product matrix $SA$ (which is computed in the first step of $\mathrm{SCW}(S,A)$), each row $i = 1, \ldots, m$ is a linear combination of the rows of $A^{(i)}$ with the coefficients $\{S(i,j) : j \in I_i\}$, regardless of the remaining rows of $A$. Therefore, we can now handle every block separately and independently of the others. Our goal, for each block $A^{(i)}$, is to choose a (non-sparse) one-dimensional sketching matrix, or just a sketching vector, $s_i \in \mathbb{R}^{|I_i|}$. Our final output matrix $S$ would be obtained by assigning the entries of $s_i$ to the non-zero entries in row $i$ of $S$, for every $1, \ldots, m$.

How should we choose $s_i$? Two natural approaches come to mind. The first is to preserve as much of the Frobenius mass of $A^{(i)}$ as possible, i.e., to maximize $\|s_i^\top A^{(i)}\|_2^2$. This is attained by setting $s_i$ to be the top left-singular vector of $A^{(i)}$.[1] This is the algorithm we call **1Shot1Vec**.

The second approach is to choose a left-singular vector of $A^{(i)}$ *at random*. The main advantage of this approach over the previous one is that it endows the algorithm with *provable* guarantees on the LRA error. To get the benefits of both approaches we can naturally combine them, leading to the algorithm we call **1Shot2Vec**. Let $S_0^* \in \mathbb{R}^{2m \times n}$ be a vertical concatenation of two copies of $S_0$, meaning that each row sparsity pattern is now repeated twice. $S_0^*$ induces the same partition of $A$ into $m$ blocks as $S_0$ (despite having twice as large of a sketching dimension), but now we may choose two sketching vectors $s_i', s_i'' \in \mathbb{R}^{|I_i| \times n}$ per block instead of just one. In **1Shot2Vec**, we set $s_i'$ to be the top left-singular vector of $A^{(i)}$, and $s_i''$ to be a random left-singular vector from the remaining ones, chosen with probability proportional to its squared corresponding singular value. The algorithm in summarized in Algorithm 2.

These two approaches to choosing $s_i$ can be grounded in formal analysis. In brief, the rows of $A$ can be partitioned into *heavy* and *light* according to their squared Euclidean norm relative to $\|A\|_F^2$. The top singular vectors of each block help preserve the heavy rows, and the random singular vectors of each block help preserve the light rows. A formal statement is given below in Theorem 2.1. A more detailed overview of the analysis is given in Section 2.3, and a formal proof appears in the appendix.

**Consistency.** We call a data-driven LRA algorithm *consistent* if a sketching matrix $S$ computed from a data matrix $A$ indeed yields a good LRA when applied to $A$. This is a natural desideratum for data-driven LRA algorithms, guaranteeing success in the naïve case where the test matrix is identical to the data matrix. The next theorem states that **1Shot2Vec** is consistent if we apply it twice, with two similar but different sketching dimensions. (This duplication is due to technical reasons in the proof — see Section 2.3 — and is not needed in practice.)

**Theorem 2.1.** *Let $m \geq \mathrm{poly}(k \log(n)/\epsilon)$ and $m' = O(\mathrm{poly}(k \log(n)/\epsilon) \cdot m)$. Let $S \in \mathbb{R}^{(m+m') \times n}$ be given by concatenating the sketching matrices computed by two calls to **1Shot2Vec**, once with $m$ and once with $m'$, on a given input matrix $A \in \mathbb{R}^{n \times d}$. Then, with probability $0.99$ over the random choices of **1Shot2Vec**, we have:[2] $\|A - \mathrm{SCW}(S,A)\|_F^2 \leq \|A - A_k\|_F^2 + \epsilon \|A\|_F^2$.*

---

[1] Scaling a row of $S$ by a constant does not change the SCW output, so we may assume $s_i$ has unit length.

[2] We remark that the guarantee of Theorem 2.1 is often referred to in the low rank approximation literature as an *additive error* LRA guarantee. The stronger *relative error* guarantee, $\|A - \mathrm{SCW}(S,A)\|_F^2 \leq (1 + \epsilon)\|A - A_k\|_F^2$ (see Theorem 2.2), in fact does not hold for **1Shot2Vec**.

**Algorithm 2:** Closed-form computation of a sketching matrix from a training matrix

**Input:** Input matrix $A \in \mathbb{R}^{n \times d}$, sketching dimension $m$.
**Output:** Sketching matrix $S$.

1 Initialize $S', S'' \in \mathbb{R}^{m \times n}$ as zero matrices
2 Choose $\xi_1, \ldots, \xi_n \in \{1, \ldots, m\}$ independently and uniformly at random
3 **for** $i = 1, \ldots, m$ **do**
4      $I_i = \{j : \xi_j = i\}$
5      $A^{(i)} \leftarrow$ restriction of $A$ to rows in $I_i$
6      $U \Sigma V^T \leftarrow$ SVD of $A^{(i)}$
7      Denote by $u_1, \ldots, u_r$ the columns of $U$ and by $\sigma_1, \ldots, \sigma_r$ the diagonal entries of $\Sigma$
8      Choose $\hat{u}$ from $\{u_2, \ldots, u_r\}$ at random with probability proportional to $\sigma_2^2, \ldots, \sigma_r^2$
9      $S'[i, I_i] \leftarrow u_1^T$
10      $S''[i, I_i] \leftarrow \hat{u}^T$            */* Note that $u_1$ and $\hat{u}$ are column vectors in $\mathbb{R}^{|I_i|}$ */
11 **return** $S'$ for **1Shot1Vec** or the vertical concatenation of $S', S''$ for **1Shot2Vec**

## 2.2 Few-Shot SGD Algorithm

Our next algorithm, **FewShotSGD**, shares with IVY the following counts — we learn the non-zero entries of the sketching matrix by optimizing a certain loss function with SGD, and we use the autograd mechanism to compute gradients. However, instead of minimizing the empirical training loss (Equation (2)), we use a different loss function, motivated by the theoretical analysis of oblivious fast LRA algorithms. This loss function is considerably faster to optimize than Equation (2), particularly since it does not require a differentiable implementation of the SVD, allowing us to use faster non-differentiable implementations. Furthermore, empirically, optimizing this loss with SGD arrives at a local minimum after just 2 or 3 iterations, allowing us to use only that many random matrices from the training set.

To specify the loss function, let $\mathcal{U}$ denote the set of all left-SVD factor matrices of our training set:

$$\mathcal{U} = \{U : A = U \Sigma V^\top \text{ is the SVD of some } A \in \mathcal{A}\}.$$

Recall that $k$ denotes the target low rank of the LRA algorithm. Let $U_k \in \mathbb{R}^{n \times k}$ denote the submatrix of $U$ that contains its first $k$ columns. Recall that the matrices in $\mathcal{A}$ are of order $n \times d$ where $k \ll d \leq n$. Let $I_0 \in \mathbb{R}^{k \times d}$ denote the result of augmenting the identity matrix of order $k$ with $d - k$ additional zero columns on the right. The loss we minimize in **FewShotSGD** is:

$$\sum_{U \in \mathcal{U}} \|U_k^\top S^\top S U - I_0\|_F^2, \tag{3}$$

where the variables are the non-zero entries of $S \in \mathbb{R}^{m \times n}$ (whose locations are given by $S_0$). This loss function is motivated by the analysis of prior LRA algorithms that use random sketching matrices, see the last paragraph of Section 2.3 for further explanation.

To minimize Equation (3) with SGD, in each round we pick a uniformly random $A \in \mathcal{A}$, compute its SVD to obtain $U$, and update $S$ according to the gradient of $\|U_k^\top S^\top S U - I_0\|_F^2$ at the current setting of $S$. Importantly, unlike IVY [33], our SVD implementation need not be differentiable. The reason IVY requires differentiable SVD is that in each iteration, due to the term $\text{SCW}(S, A)$ in eq. (2), it computes a (partial) SVD of $SA$ and of $A\tilde{V}$ — both of which involve variables in $S$. Our loss computes the SVD of $A$, which is independent of $S$, and thus no gradients flow through it.

The motivation behind the choice of loss function in Equation (3) is the following theorem, which can be viewed as a statement of consistency for **FewShotSGD**: If the loss is small, then SCW provably computes a good low rank approximation of $A$ using $S$.

**Theorem 2.2.** *Let $A = U \Sigma V^\top$ be a matrix with its SVD. For a sketching matrix $S \in \mathbb{R}^{m \times n}$, let $\epsilon = \|U_k^\top S^\top S U - I_0\|_F^2$. Then, $\|A - \text{SCW}(S, A)\|_F^2 \leq (1 + O(\epsilon)) \cdot \|A - A_k\|_F^2$.*

As a result, the loss minimized in eq. (3) is directly related to the empirical loss (eq. (2)).

## 2.3  Overview of Analysis

The proofs of Theorems 2.1 and 2.2 are quite involved and appear in full in the appendix. In this section we give an intuitive overview of the main ideas in their analysis.

**One-shot algorithm: Theorem 2.1.** Recall that **1Shot2Vec** partitions the rows of $A$ into $m$ blocks according to $S_0$, and takes the top left-singular vector and a random left-singular vector from each block. Let $\epsilon' = \text{poly}(\epsilon/k \log n)$. To analyze the algorithm, we classify the rows $a_1, ..., a_n$ of $A$ into three types, according to their relative *mass* $\|a_i\|_2^2/\|A\|_F^2$ (note that the masses sum to 1):

- Heavy rows, whose mass is at least $1/(\epsilon'm)$.
- Light rows, whose mass is at most $\epsilon'/m$.
- Medium rows, whose mass is between $\epsilon'/m$ and $1/(\epsilon'm)$.

The mass of each top-$k$ direction of $A$, which we aim to preserve in the sketch, is distributed somehow over the rows of $A$. We show how the mass is recovered by the sketch for each type of row.

Consider a heavy row $a_i$. Since each heavy row has mass at least $1/(\epsilon'm)$, there can be at most $\epsilon'm$ of them. Since **1Shot2Vec** partitions the rows of $A$ into $m$ blocks at random, by standard hashing arguments, with high probability there will be no collisions, and thus $a_i$ would be the only heavy row in its block. Since its mass is significantly larger than the non-heavy rows, it can be shown that it is highly correlated with the top-singular vector of the block, which the algorithm takes into the sketch. Thus, $a_i$ is well-represented in the sketch.

Next, consider the light rows. Here we use the notion of *projection-cost preserving sketches* (PCPs) [23, 18, 14, 45], and specifically, the additive-error PCPs introduced by Bakshi and Woodruff [7]. A $(k, \epsilon)$-PCP of $A$ is a matrix $B$ (with many fewer rows) that approximately preserves the Frobenius norm of $A$ under every projection away from a $k$-dimensional subspace. Namely, $\|B - BP\|_F^2 = \|A - AP\|_F^2 \pm \epsilon\|A\|_F^2$ for every rank-$k$ projection $P$. The connection to LRA is that the LRA problem on $A$ is equivalent to $\text{argmin}_P \|A - AP\|_F^2$, and the PCP property implies that we can find an approximate LRA for $A$ by solving LRA on the (much smaller) matrix $B$.

In [7], it is shown that a $(k, \epsilon)$-PCP of $A$ can be computed with high probability by sampling each row $a_i$ of $A$ with probability at least $(1/\epsilon')\|a_i\|_2^2/\|A\|_F^2$. For light rows as defined above, this bound is at most $1/m$. Since **1Shot2Vec** partitions the rows of $A$ into blocks uniformly at random, each block contains each light row with probability $1/m$, and thus forms a $(k, \epsilon)$-PCP for the light rows of $A$ with high probability. As a result, the light rows are well-represented in the sketch.

Finally, we handle the medium rows. Here we make use of the fact that Theorem 2.1 invokes **1Shot2Vec** twice, once with sketching dimension $m$ and once with $m' = m/\epsilon'^2$, and concatenates the resulting sketches. If a row $a_i$ is medium in the first invocation, then its mass is at least $\epsilon'/m = 1/(\epsilon'm')$, rendering it heavy in the second invocation. Thus, row $a_i$ is well-represented in at least one of the two sketches. Overall, the fact that every row is well-represented in the sketch implies that each of the top-$k$ directions is approximately preserved in the resulting LRA, as needed.

**Few-shot algorithm: Theorem 2.2.** The theorem is based on the original analysis of the data-oblivious SCW algorithm in [15, 16]. The main point is that the loss can be written as the sum of contributions of the first $k$ columns and the last $d - k$ columns, as follows:

$$\epsilon = \|U_k^\top S^\top SU - I_0\|_F^2 = \|U_k^\top S^\top SU_k - I_k\|_F^2 + \|U_k^\top S^\top SU_{d-k} - \mathbf{0}\|_F^2, \tag{4}$$

where $I_k$ is the identity of order $k$, $U_{d-k}$ is $U$ without its top-$k$ columns, and $\mathbf{0}$ is the all-zero matrix of order $k \times (d - k)$. The first term, $\|U_k^\top S^\top SU_k - I\|_F^2$, measures how well the sketch preserves the top-$k$ space of $A$ (note that $U_k^\top U_k = I$). The second term, $\|U_k^\top S^\top SU_{d-k} - \mathbf{0}\|_F^2$, measures how well the sketch keeps the top-$k$ space of $A$ orthogonal to its bottom-$(d - k)$ space (note that $U_k^\top U_{d-k} = \mathbf{0}$). By eq. (4), each of the terms is bounded by $\epsilon$, which is sufficient for low rank approximation with relative error $O(\epsilon)$.

# 3  Experiments

We evaluate our algorithms empirically and compare them to previous work on data-driven LRA. For a direct comparison, we use three datasets from [33], with the same train/test partitions — Logo,

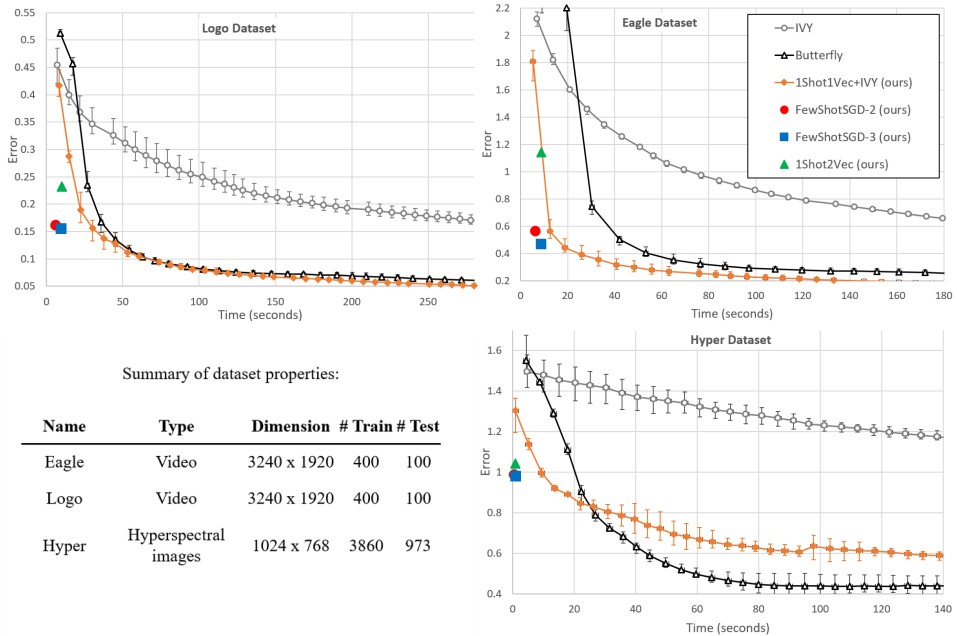

Figure 1: Test set error per training time, with target rank $k = 10$ and sketching dimension $m = 40$.

Eagle and Hyper [32]. All are publicly available. The performance of a sketching matrix $S$ on a set of input matrices $\mathcal{A}$ is measured similarly to [33], as

$$\mathrm{err}_{\mathcal{A}}(S) := \frac{1}{|\mathcal{A}|} \sum_{A \in \mathcal{A}} \left( \|A - \mathrm{SCW}(S, A)\|_F^2 - \|A - A_k\|_F^2 \right),$$

where we recall $A_k$ denotes the optimal rank-$k$ approximation of $A$. This measures the additive gap between the sketch-based LRA and the optimal LRA of each matrix, averaged over the test set matrices. Experiments were run on an NVIDIA GeForce RTX 2080 TI graphics card.

**Our algorithms.** To test our few-shot algorithms, we evaluate **1Shot2Vec** with one randomly chosen training matrix, and **FewShotSGD** with either 2 or 3 randomly chosen training matrices (that is, 2 or 3 steps of SGD). In addition, we incorporate our approach into a many-shot algorithm, by initializing the sketching matrix of IVY using our method **1Shot1Vec** instead of a random sign matrix.

**Baselines.** As baselines, we evaluate three methods: IVY [33], Butterfly [3], and Greedy [38].[3]

*Butterfly.* For an integer $k > 0$, a butterfly gadget is a matrix $B \in \mathbb{R}^{2^k \times 2^k}$ formed as the product of $k$ sparse matrices, $B = \prod_{i=0}^{k-1} B_i$, where $B_i \in \mathbb{R}^{2^k \times 2^k}$ has 1 in entry $(j, j')$ if either $j = j'$ or $j \oplus 2^{k-i} = j'$ (where $\oplus$ denotes XOR), and 0 otherwise. These well-known gadgets arise in the context of the Fast Fourier Transform. Ailon et al. [3] suggest using them for learned LRA, by forming a sketching matrix from $m$ rows of $B$, and learning new values for the non-zero entries of the $B_i$'s by minimizing the empirical loss with SGD, similarly to IVY.

*Greedy.* Liu et al. [38] suggest initializing the sketching matrix by the following greedy procedure. Choose a set $V$ of 10 uniformly random values in $[-2, 2]$. Starting with a zero matrix $S_0 \in \mathbb{R}^{m \times n}$, we fill in column by column. In step $i = 1, \ldots, n$, let $S_{i,j,v}$ denote the matrix obtained from $S_{i-1}$ by putting $v$ in entry $(j, i)$. Choose $S_i$ as the best performing matrix among $\{S_{i,j,v} : j \in \{1, \ldots, m\}, v \in V\}$ over some subset of the training set (say, a single random matrix).

**Training times.** We measure the error per training time for each method. The results are reported in Figure 1 and in Table 1. We do not include Greedy in these results, since its training time is much longer than the other methods (35–40 minutes per matrix, versus up to 4 minutes for the other

---

[3]We remark that [38] also proposed a four-sketch alternative to IVY, based on [6], which has faster asymptotic running time. However, they reported that it falls short of IVY empirically in its attained error.

methods), due to the sequential grid search it performs. We report its performance later in this section, where we measure the error per number of sampled training matrices.

The experiments show that our few-shot algorithms, **1Shot2Vec** and **FewShotSGD**, achieve low error much faster than IVY and Butterfly, on all datasets. Table 1 lists the time it takes the baselines to reach the same error. Note that **1Shot2Vec** is somewhat slower and less accurate than **Few-ShotSGD** with 2 iterations; nonetheless, it has the advantage of being a "closed-form" algorithm that does not perform learning at all and does not require automatic gradient computation.

If IVY and Butterfly are allowed to run longer, and use sufficiently many training matrices, they eventually attain lower error than **1Shot2Vec** and **FewShotSGD**. This is our motivation for introducing **1Shot1Vec+IVY** (which we recall is the initialization of IVY with **1Shot1Vec**), as a way to use our techniques to also improve the many-shot algorithms. This algorithm significantly outperforms IVY, and outperforms Butterfly on Logo, Eagle, and the small time/sample regime of Hyper.

Note that while the initial error of **1Shot1Vec+IVY** is similar to IVY, the former error improves much faster in subsequent iterations. This is related to the formal guarantee of the oblivious SCW initialization, by which almost all of its initial $\pm 1$ choices hit its target loss, irrespective of their location with respect to the local minima of IVY. There are $\exp(n)$ such initial choices even for a fixed sparsity pattern. Our deterministic data-tailored initialization, on the other hand, is better positioned to choose a point which, while having similar loss to SCW, is nearer to a local minimum.

*Remark.* In light of the strong performance of **1Shot1Vec+IVY**, it may seem natural to apply the same initialization to the Butterfly. This turns out to not be possible. The trained parameters of Butterfly are not entries of $S$ as in IVY, but implicit parameters that make up $S$ by multiplying a sequence of sparse matrices. Since a general sketching matrix is specified by $nm$ parameters, whereas a butterfly gadget truncated to $m$ rows is specified by only $O(n \log m)$ parameters [4, 3], arbitrary initializations of the sketching matrix cannot be realized by the butterfly gadget.

**Testing times.** Next we report the running times of the testing phase on each dataset, to substantiate the claimed advantage of sketch-based approximate LRA over exact LRA by SVD. Note that all of the algorithms evaluated above (IVY, Butterfly and our methods) have the same runtime in the test phase, since in this phase they all perform the same SCW procedure (Algorithm 1).

We measure the total runtime of computing LRA on the test matrices by SVD versus sketch-based LRA. On Eagle / Logo / Hyper, LRA by SVD takes **88.8s / 89.7s / 190.7s**, while sketch-based LRA takes **6.6s / 7.0s / 22.1s**, respectively. The average speed-up is by a multiplicative factor of **11.6**.

**Sample complexity.** Finally, we measure the error achieved by each method when given access to only a small number of random matrices from the dataset (i.e., we evaluate each method in the few-shot setting). The results are reported in Table 2. They show that our methods obtain better performance than the baselines, with the exception of Greedy, which achieves lower errors on two of the datasets, even with just one sample matrix. However, we note that the training time of Greedy is a few orders of magnitude larger: 2306.4 seconds on Logo, 2325.5 seconds on Eagle, and 953.1 seconds on Hyper, for a single training matrix. In contrast, our few-shot methods train for up to 10 seconds on up to 3 matrices (on either dataset), and the many-shots methods train for up to 30 seconds. Determining whether it is possible to obtain fast algorithms that also learn the sparsity pattern is an important direction for future work.

**Additional experiments.** In the experiments above, our learned sketching matrices have only one non-zero entry per column. Recent work has suggested that using more non-zero entries per column (up to 8) can be more beneficial [52, 42]. The appendix includes some experimental results with sparsity 8.

It is also possible to consider learning the sketching matrix on one dataset and applying it to another (transfer learning), or to learn it jointly from two different datasets (mixed learning). Transfer learning in the context of LRA has been recently considered in [48]. In the appendix we include some preliminary result in this vein, which suggest that these approaches can be useful for some datasets.

Table 1: Training time comparison of few-shot to many-shot data driven LRA algorithms. For each of our few-shot algorithms (**1Shot2Vec** and **FewShotSGD** with 2 and 3 iterations), we report the error it attains on each dataset, and the training time it takes. Then, for each of the many shot algorithms (IVY, Butterfly and **1Shot1Vec**+IVY), we report the training time it takes to achieve the same error. N/A means the algorithm did not attain the error in the time allotted to the experiment.

| Few-shot algorithm | Dataset | Few-shot error | Time | Time to error for many-shot algorithms | | |
| --- | --- | --- | --- | --- | --- | --- |
| | | | | IVY | Butterfly | 1Shot1Vec+IVY |
| 1Shot2Vec | Eagle | 1.14 | 8.892s | 56.56s | 30.34s | 12.47s |
| | Logo | 0.232 | 10.06s | 143.75s | 26.86s | 22.43s |
| | Hyper | 1.04 | 0.78s | N/A | 22.35s | 9.36s |
| FewShotSGD-2 | Eagle | 0.563 | 6.417s | N/A | 42.09s | 18.81s |
| | Logo | 0.161 | 6.17s | N/A | 45.14s | 30.16s |
| | Hyper | 0.99 | 0.5s | N/A | 22.35s | 13.66s |
| FewShotSGD-3 | Eagle | 0.467 | 8.879s | N/A | 53.26s | 25.05s |
| | Logo | 0.155 | 9.65s | N/A | 45.14s | 30.16s |
| | Hyper | 0.98 | 0.99s | N/A | 22.35s | 13.66s |

Table 2: Error attained by each method on the test set of each dataset, after training on 1, 2 and 3 random matrices from the training set.

| # Matrices | Dataset | 1Shot2Vec | FewShotSGD | 1Shot1Vec+IVY | IVY | Butterfly | Greedy |
| --- | --- | --- | --- | --- | --- | --- | --- |
| 1 | Eagle | 1.14 | – | 1.888 | 2.172 | 2.163 | 0.062 |
| | Logo | 0.232 | – | 0.397 | 0.485 | 0.51 | 0.23 |
| | Hyper | 1.04 | – | 1.241 | 1.573 | 1.491 | 0.364 |
| 2 | Eagle | – | 0.563 | 0.65 | 1.864 | 2.036 | 0.047 |
| | Logo | – | 0.161 | 0.289 | 0.427 | 0.468 | 0.263 |
| | Hyper | – | 0.99 | 1.106 | 1.559 | 1.444 | 0.393 |
| 3 | Eagle | – | 0.467 | 0.507 | 1.617 | 0.785 | 0.064 |
| | Logo | – | 0.155 | 0.165 | 0.398 | 0.223 | 0.371 |
| | Hyper | – | 0.98 | 1.015 | 1.538 | 1.271 | 0.367 |

# Acknowledgments and Disclosure of Funding

We thank the anonymous reviewers for their useful suggestions. This research was supported in part by the NSF TRIPODS program (awards CCF-1740751 and DMS-2022448), NSF awards CCF-2006798 and CCF-1815840, Office of Naval Research (ONR) grant N00014-18-1-2562, and Simons Investigator Awards.

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
