# Few-Shot Data-Driven Algorithms for Low Rank Approximation: Supplementary Material

**Piotr Indyk**
MIT
indyk@mit.edu

**Tal Wagner**
Microsoft Research Redmond
tal.wagner@gmail.com

**David P. Woodruff**
Carnegie Mellon University
dwoodruf@cs.cmu.edu

# A Proof of Theorem 2.1

Let us restate the theorem. Recapping notation, let $A \in \mathbb{R}^{n \times d}$ be an input matrix. Let $a_1, \ldots, a_n$ denote its rows. Let $k$ be the target low rank, and let $\epsilon > 0$ be an error parameter. Let $S_0 \in \mathbb{R}^{m \times n}$ be a random matrix in which every column is chosen uniformly at random from the standard basis $e_1, \ldots, e_m$. $S_0$ induces a partition of the rows $A$ into $m$ blocks, $B_1, \ldots, B_m$, where $B_j$ is comprised of the rows $\{a_i : S(j, i) = 1\}$. Let $\beta_{j,1} \geq \ldots \geq \beta_{j,d}$ denote its singular values, with corresponding left-singular vectors $w_{j,1}, \ldots, w_{j,d}$. We sample one left-singular vector $w_{j,\ell}$ with probability $\beta_{j,\ell}^2 / \|B_j\|_F^2$, and replace the 1-entries in the $j$th row of $S_0$ with the entries of $w_{j,\ell}$. Doing this for each $B_j$ results in our sketching matrix $S$.

We do this with two random matrices $S_0' \in \mathbb{R}^{m' \times n}$ and $S_0'' \in \mathbb{R}^{m'' \times n}$, where $m', m''$ are both $poly(k, \epsilon^{-1}, \log n)$, and are within a low order term of each other. The final sketching matrix $S$ is the vertical concatenation of $S'$ and $S''$, with sketching dimension $m = m' + m''$. The claim is that,

**Theorem A.1** (restatement of Theorem 2.1 from main text). *If $m = poly(k, \epsilon^{-1}, \log n)$, then with constant probability (say $0.99$), $\|A - SCW(S, A)\|_F^2 \leq \|A - A_k\|_F^2 + \epsilon\|A\|_F^2$.*

In the remainder of this section we prove Theorem A.1.

## A.1 Preliminaries

Let $A = U\Sigma V^T$ denote the SVD of $A$. The singular values of $A$ (i.e., the diagonal entries of $\Sigma$) are denoted $\sigma_1 \geq \ldots \geq \sigma_d$. The corresponding left-singular vectors (columns of $U$) are $u_1, \ldots, u_d$, and the corresponding right-singular vectors (columns of $V$) are $v_1, \ldots, v_d$. It will be convenient to denote by $\epsilon_1, \epsilon_2, \epsilon_3 \ldots$ sizes that are all $poly(\epsilon, 1/k, 1/\log(n))$.

First, we recall a basic lemma about SCW, which appears as Lemma 44 in [7]:

**Lemma A.2** (Lemma 44 in [7]). $SCW(S, A)$ *returns the best rank-$k$ approximation of $A$ (in Frobenius norm) within the row-space of $SA$. Put otherwise, it returns the best rank-$k$ approximation of $A$ that can be written as $BSA$, where $B \in \mathbb{R}^{n \times m}$ is any matrix of rank $k$.*

Therefore, our goal is to show that this row-space contains a sufficiently large component of each of $v_1^T, \ldots, v_k^T$ (the top-$k$ right-singular vectors of $A$, written as row vectors). To reason about the row-space of $SA$, we have the following simple observation: whenever the algorithm samples a left-singular vector of a block in forming $S$, it adds the corresponding right-singular vector to the row-space of $SA$.

**Observation A.3.** *Let $B_j$ be a block with singlular values $\beta_1 \geq \ldots \geq \beta_d$, left-singular vectors $w_1, \ldots, w_d$, and right-singular values $y_1, \ldots, y_d$. If the algorithm samples $w_\ell$ for the jth row of S, then the row-space of $SA$ contains $y_\ell$.*

*Proof.* It can be easily seen that the $j$th row of $SA$ equals $w_\ell B_j$, which equals $\beta_\ell \cdot y_\ell$. $\qquad\square$

Let $k' \leq k$ the largest index such that $\sigma_{k'} \geq (\epsilon/k)\|A\|_F^2$. It suffices for us to find an approximate rank-$k'$ approximation of $A$. Indeed, if $A'$ is a rank-$k'$ approximation of $A$, meaning it satisfies

$$\|A - A'\|_F^2 \leq \|A - A_{k'}\|_F^2 + \epsilon\|A\|_F^2,$$

then it is also a good rank-$k$ approximation up to scaling $\epsilon$ by a constant, since

$$\|A - A_{k'}\|_F^2 = \|A - A_k\|_F^2 + \sum_{i=k'+1}^{k} \sigma_i^2 \leq \|A - A_k\|_F^2 + \epsilon\|A\|_F^2,$$

and therefore

$$\|A - A'\|_F^2 \leq \|A - A_k\|_F^2 + 2\epsilon\|A\|_F^2.$$

To simplify notation, we will assume that $k = k'$, and

$$\sigma_i^2 \geq \frac{\epsilon}{k}\|A\|_F^2 \quad \text{for all } i = 1, \ldots, k. \tag{1}$$

This is the most difficult case of the above ($k' \leq k$) and does not limit the generality of the proof.

## A.2 Row Classification

We classify the rows $a_1, \ldots, a_n$ into light, medium and heavy as follows. We define a row $a_i$ as *light* if it satisfies,

$$\frac{1}{\epsilon_2} \cdot \frac{\log n}{\epsilon_4^2} \cdot \frac{\|a_i\|_2^2}{\|A\|_F^2} \leq \frac{1}{m}. \tag{2}$$

A row $a_i$ is defined as heavy if

$$\frac{\|a_i\|_2^2}{\|A\|_F^2} \geq \frac{1}{\epsilon_5 m}. \tag{3}$$

If a row satisfies neither eq. (2) nor eq. (3), it is defined as *medium*.

Let $A^L, A^M, A^H$ be a partition of $A$ into submatrices that contain the light, medium and heavy rows respectively. Observe that for every singular value $\sigma_j$ we have

$$\sigma_j^2 = \|Av_j\|_2^2 = \|A^L v_j\|_2^2 + \|A^M v_j\|_2^2 + \|A^H v_j\|_2^2. \tag{4}$$

For every block $B$, we similarly partition it into three matrices $B^L, B^M, B^H$ of the light, medium and heavy (respectively) rows assigned to it.

Next we prove some useful properties of each class of rows.

### A.2.1 Heavy Rows

**Lemma A.4.** *The number of heavy rows is at most $\epsilon_5 m$. Furthermore, with high probability every block contains at most one heavy row.*

*Proof.* Let $n_H$ be the number of heavy rows. Since each heavy row $a_i$ satisfies eq. (3), the sum of their squared norms is at least $n_H \cdot \|A\|_F^2/(\epsilon_5 m)$. On the other hand, the sum of their squared norms equals $\|A^H\|_F^2$, which is at most $\|A\|_F^2$. Thus $n_H \leq \epsilon_5 m$ The second part of the lemma follows since the rows are hashed into $m$ buckets uniformly at random, and $\epsilon_5 \ll 1$. $\qquad\square$

### A.2.2 Medium Rows

**Lemma A.5.** *With high probability, for every block $B$ it holds that $\|B^M\|_F^2 \leq \epsilon_6 \|A\|_F^2$, where $\epsilon_6 = \log(n)/(m\epsilon_5\epsilon_2\epsilon_4^2)$.*

*Proof.* Let $n_M$ be the total number of medium rows in $A$. We upper bound $n_M$. On the one hand, since every medium row does not satisfy eq. (2), we have

$$\|A^M\|_F^2 = \sum_{a_i \in A^M} \|a_i\|_2^2 \geq n_M \cdot \frac{\epsilon_2\epsilon_4^2}{m\log n} \cdot \|A\|_F^2.$$

On the other hand, $\|A^M\|_F^2 \leq \|A\|_F^2$. Together,

$$n_M \leq \frac{m\log n}{\epsilon_2\epsilon_4^2}.$$

The medium rows are assigned to the $m$ blocks at random, so by standard balls-into-bins bounds, the maximum number of medium rows in any block is with high probability at most $2\max\{n_M/m, \log m\}$. Since $m \leq n$, both terms in the max are upper bounded by $\log(n)/(\epsilon_2\epsilon_4^2)$, so the maximum number of medium rows in any block is at most $2\log(n)/(\epsilon_2\epsilon_4^2)$. By the definition of medium rows, the squared norm of each is at most $\|A\|_F^2/(\epsilon_5 m)$, and the lemma follows. $\qquad\square$

### A.2.3 Light Rows

We will use the notion of projection-cost preserving sketches (PCPs) [5, 3, 2, 6]. We use the additive error variant, introduced in [1] (whereas the former mentioned works used the relative error variant).

**Definition A.6.** *A matrix $B$ with the same column dimension as $A$ is called a $(k, \epsilon)$-projection cost preserving sketch (abbrev. $(k, \epsilon)$-PCP) of $A$ if for every orthogonal projection $P$ onto a rank-$k$ space, it holds that*

$$\|B(I - P)\|_F^2 = \|A(I - P)\|_F^2 \pm \epsilon\|A\|_F^2.$$

The following theorem from [1] shows that a $(k, \epsilon)$-PCP of a given $n$-row matrix can be constructed using importance sampling over the rows, proportionally to their masses, as long as they are over-sampled by a factor of $\Omega(k^2 \log(n)/\epsilon^2)$.

**Theorem A.7** ([1]). *There is a universal constant $c > 0$ such that the following holds. Let $A$ be a given matrix with $n$ rows $a_1, \ldots, a_n$. Let $p_1, \ldots, p_n$ be a sequence that satisfies*

$$p_i \geq \min\left\{1, \frac{ck^2 \log(n/\delta)}{\epsilon^2} \cdot \frac{\|a_i\|_2^2}{\|A\|_F^2}\right\}.$$

*Let $B$ be a matrix that includes each row $a_i$, multiplied by $1/\sqrt{p_i}$, with independent probability $p_i$. Then, with probability $1 - \delta$, $B$ is a $(k, \epsilon)$-PCP of $A$.*

As a consequence, in our setting we have the following.

**Corollary A.8.** *Suppose $\|A^L\|_F^2 \geq \frac{1}{3c}\epsilon_2\|A\|_F^2$, where $c$ is the constant from Theorem A.7. Then, with high probability, for every block $B$ it holds that $\sqrt{m} \cdot B^L$ is a $(1, \epsilon_4)$-PCP of $A^L$.*

*Proof.* Let $B$ be a block. Let $a_i$ be a light row of $A$. Using eq. (2) and the assumption $\|A^L\|_F^2 \geq \frac{1}{3c}\epsilon_2\|A\|_F^2$, we have,

$$\frac{c\log(n^3)}{\epsilon_4^2} \cdot \frac{\|a_i\|_2^2}{\|A^L\|_F^2} \leq \frac{1}{\epsilon_2} \cdot \frac{\log n}{\epsilon_4^2} \cdot \frac{\|a_i\|_2^2}{\|A\|_F^2} \leq \frac{1}{m}.$$

Since each row of $A^L$ is included in $B^L$ with probability $1/m$, we can view $B^L$ as constructed by Theorem A.7 with $p_i = 1/m$, for a $(1, \epsilon_4)$-PCP with failure probability $\delta = 1/n^2$. Therefore, $\sqrt{m} \cdot B^L$ is a $(1, \epsilon_4)$-PCP of $A^L$ with probability $1 - 1/n^2$. The corollary follows by a union bound over all $m \leq n$ blocks. $\qquad\square$

Let us make the following small point about PCPs with additive error:

**Claim A.9.** *Suppose $B$ is a $(1, \epsilon_4)$-PCP of $A$. Then, for every unit vector $x$, we have $\|Bx\|_2^2 = \|Ax\|_2^2 \pm 2\epsilon_4 \|A\|_F^2$.*

*Proof.* The PCP property implies that
$$\|B(I - xx^T)\|_F^2 = \|A(I - xx^T)\|_F^2 \pm \epsilon_4 \|A\|_F^2.$$
Applying the Pythagorean identity to both sides,
$$\|B\|_F^2 - \|Bxx^T\|_F^2 = \|A\|_F^2 - \|Axx^T\|_F^2 \pm \epsilon_4 \|A\|_F^2.$$
The PCP property also implies $\|B\|_F^2 = (1 \pm \epsilon_4)\|A\|_F^2$, and plugging this above an rearranging yields
$$\|Bxx^T\|_F^2 = \|Axx^T\|_F^2 \pm 2\epsilon_4 \|A\|_F^2.$$
The claim follows since $\|Cxx^T\|_F^2 = \|Cx\|_2^2$ for every matrix $C$ and unit direction $x$. $\square$

## A.3 Recovering Top Directions

We call a block $B$ *heavy* if it contains a heavy row, and *light* otherwise (so light blocks may contain both light and medium rows). For a heavy row $a$, let $B_a$ denote the heavy block that contains it. Let $w_a, \beta_a, y_a$ denote its top left-singular vector, top singular value, and top right-singular vector respectively.

The following lemma shows that for every heavy row $a$ which is sufficiently well-correlated with some $v_i$ (a top-$k$ direction of $A$), the row space of the sketched matrix $SA$ contains a vector ($y_a$) highly correlated with $a$ (and thus well-correlated with $v_i$).

**Lemma A.10** (recovery from heavy rows). *With high probability, the following holds for an appropriate choice of $\epsilon_7$ (cf. eq. (5)). For every heavy row $a$, for which there is $i \in \{1, \dots, k\}$ such that $(a^T v_i)^2 \geq \epsilon_7 \sigma_i^2$, the row-space of $SA$ contains $y_a$. Furthermore, $\frac{1}{\|a\|} \cdot |y_a^T a| \geq 1 - \epsilon_9$.*

*Proof.* We start by lower-bounding the quantity $\|a\|_2^2 / \|B_a\|_F^2$. We can break up $\|B_a\|_F^2$ into heavy, medium and light rows, $\|B_a\|_F^2 = \|B_a^H\|_F^2 + \|B_a^M\|_F^2 + \|B_a^L\|_F^2$. By Lemma A.4, $a$ is the only heavy row in $B_a$, so $\|B_a^H\|_F^2 = \|a\|_2^2$. Furthermore,
$$\|a\|_2^2 \geq (a^T v_i)^2 \geq \epsilon_7 \sigma_i^2 \geq \epsilon_7 \cdot \frac{\epsilon}{k} \|A\|_F^2,$$
having used eq. (1) for the last inequality. By Lemma A.5, $\|B_a^M\|_F^2 \leq \epsilon_6 \|A\|_F^2$. By Corollary A.8, $\|B_a^L\|_F^2 \leq \frac{1}{m}(1 + \epsilon_4)\|A^L\|_F^2 \leq \frac{1}{m}(1 + \epsilon_4)\|A\|_F^2$. Putting all the bounds together,
$$\frac{\|a\|_2^2}{\|B_a\|_F^2} = 1 - \frac{1}{\frac{\|a\|_2^2}{\|B_a^M\|_F^2 + \|B_a^L\|_F^2} + 1} \geq 1 - \frac{1}{\frac{(\epsilon_7 \cdot (\epsilon/k)\|A\|_F^2}{(\epsilon_6 + \frac{1}{m}(1 + \epsilon_4))\|A\|_F^2} + 1}.$$
By choosing
$$\epsilon_7 \gg \frac{k}{\epsilon} \cdot \left( \epsilon_6 + \frac{1}{m}(1 + \epsilon_4) \right), \tag{5}$$
we get $\|a\|_2^2 \geq (1 - \epsilon_8)\|B_a\|_F^2$, for $\epsilon_8$ of our choice. Since $a$ is a row in $B_a$, we have $\|B_a\|_2^2 \geq \|a\|_2^2$, and therefore,
$$\beta_a^2 = \|B_a\|_2^2 \geq \|a\|_2^2 \geq (1 - \epsilon_8)\|B_a\|_F^2.$$
We sample $w_a$ from $B_a$ with probability $\beta_a^2 / \|B_a\|_F^2$, which is at least $1 - \epsilon_8$. By Observation A.3, this implies that the row-space of $SA$ contains $y_a$ with that probability. By Lemma A.4 there are at most $\epsilon_5 m$ heavy rows and thus at most that many heavy blocks, so we choose $\epsilon_8 \ll 1/(\epsilon_5 m)$, and we can take a union bound and ensure the above happens with high probability for all heavy blocks simultaneously.

For the second part of the lemma, let $e_a$ denote the standard basis vector corresponding to the index of row $a$ in $B_a$, so that $a^T = e_a^T B_a$. Since $\|e_a^T B_a\|_2^2 \geq (1 - \epsilon_8)\|B_a\|_F^2$, it is not hard to see that $|w_a^T e_a| \geq 1 - 3\epsilon_8$ by decomposing $e_a$ over the orthonormal basis of left-singular vectors of $B_a$. But $w_a = \beta_a^{-1} B_a y_a$, and therefore
$$1 - 3\epsilon_8 \leq \beta_a^{-1} |y_a^T B_a^T e_a| = \beta_a^{-1} |y_a^T a| \leq \frac{1}{\|a\|} |y_a^T a|,$$
where the last inequality is since $\beta_a = \|B_a\|_2 \geq \|a\|$. We can take $\epsilon_9 = 3\epsilon_8$. $\square$

Let $\epsilon_3 = \epsilon_2/(3c)$, where $c$ is the constant from Theorem A.7.

**Lemma A.11** (recovery from light rows)**.** *With high probability, the following holds. For every $i \in \{1, \ldots, k\}$ such that $\|A^L v_i\|_2^2 \geq \epsilon_3 \sigma_i^2$, the projection of $v_i$ on the row-space of $SA$ has length at least $1 - \epsilon_1$.*

*Proof.* First, note that if no $i \in \{1, \ldots, k\}$ satisfies $\|A^L v_i\|^2 \geq \epsilon_3 \|A\|_F^2$ then the lemma holds trivially, whereas otherwise, the conclusion in Corollary A.8 holds. Therefore we many assume the latter henceforth.

The algorithm samples a left-singular vector from each block, thus adding the corresponding right-singular vector to the row-space of $SA$ (cf. Observation A.3). We restrict our attention to the light blocks, and visualize sampling vectors from them one by one (in an arbitrary order).

Let $W \subset \mathbb{R}^d$ denote the subspace spanned by the right-singular vectors of the light blocks sampled so far, together with all the heavy and medium rows in $A$. Let $P_W$ denote the orthogonal projection on it, and $P_W^\perp = I - P_W$ the orthogonal projection orthogonal to it. Note that $AP_W^\perp = A^L P_W^\perp$. If $\|P_W v_i\|_2^2 \geq 1 - \epsilon_1$, then we are done with $v_i$ (the conclusion of the lemma holds for it). Otherwise, $\|P_W^\perp v_i\|_2^2 > \epsilon_1$, which we assume henceforth.

We have,

$$\|A^L P_W^\perp v_i\|_2^2 = \|A P_W^\perp v_i\|_2^2 = \|\sum_j u_j \sigma_j v_j^T P_W^\perp v_i\|_2^2 = \sum_j \sigma_j^2 (v_j^T P_W^\perp v_i)^2 \geq \sigma_i^2 (v_i^T P_W^\perp v_i)^2 = \sigma_i^2 \|P_W^\perp v_i\|_2^4.$$

Letting $x = \frac{1}{\|P_W^\perp v_i\|_2} P_W^\perp v_i$ denote the unit direction along $P_W^\perp v_i$, the above together with eq. (1) implies

$$\|A^L x\|_2^2 \geq \sigma_i^2 \|P_W^\perp v_i\|_2 \geq \frac{\epsilon_1 \cdot \epsilon}{k} \|A\|_F^2.$$

Let $B$ be the next light block we sample from. By Corollary A.8, $\sqrt{m} B^L$ is a $(1, \epsilon_4)$-PCP of $A^L$, which by Claim A.9 implies

$$m \|B^L x\|_2^2 \geq \left( \frac{\epsilon_1 \cdot \epsilon}{k} - 2\epsilon_4 \right) \cdot \|A^L\|_F^2.$$

Let $\{w_j\}, \{\beta_j\}, \{y_j\}$ denote the left-singular vectors, singular values, and right-singular vectors respectively of $B$. Then, the expected projection length of $x$ along the singular vector $y_j$ we sample from $B$ into $SA$ is,

$$\sum_j \frac{\beta_j^2}{\|B\|_F^2} \cdot (y_j^T x)^2 = \frac{\|Bx\|_2^2}{\|B\|_F^2} \geq \frac{\|B^L x\|_2^2}{\|B^L\|_F^2 + \|B^M\|_F^2 + \|B^H\|_F^2}.$$

The numerator $\|B^L x\|_2^2$ is lower bounded by $\frac{1}{m}(\frac{\epsilon}{k} - 2\epsilon_4)\|A\|_F^2$ as shown above. For the denominator, we have

- $\|B^L\|_F^2 \leq \frac{1}{m}(1 + \epsilon_4)\|A\|_F^2$ by the PCP property of $B$.

- $\|B^M\|_F^2 \leq \epsilon_6 \|A\|_F^2$ by Lemma A.5.

- $B^H = 0$ since $B$ is a light block (no heavy rows).

Putting these together, the expected mass is at least

$$\frac{\frac{1}{m}(\epsilon_1 \cdot \frac{\epsilon}{k} - 2\epsilon_4)\|A\|_F^2}{\frac{1}{m}(1 + \epsilon_4)\|A\|_F^2 + \epsilon_6 \|A\|_F^2} = \frac{\epsilon_1 \cdot \frac{\epsilon}{k} - 2\epsilon_4}{1 + \epsilon_4 + m\epsilon_6},$$

which is at least say $\Omega(\epsilon_1/(\epsilon_6 m))$ with an appropriate choice of $\epsilon_4$. Since the mass is bounded between 0 and 1, by a Markov bound on 1 minus this expected mass, the sampled mass is at least $\Omega(\epsilon_1/(\epsilon_6 m))$ with probability at least $\Omega(\epsilon_1/(\epsilon_6 m))$. So, after $O(\epsilon_6 m/\epsilon_1)$ steps in expectation we gain $\Omega(\epsilon_1/(\epsilon_6 m))$ mass along the direction $x$. Recalling that $x$ is the direction of $P_W^\perp v_i$, we gain $\Omega(\epsilon_1/(\epsilon_6 m))$ mass along $v_i$, all in a direction orthogonal to what we had so far of $v_i$. To ensure the lemma, we need to repeat this up to $O(1/\epsilon_1)$ times per direction for each of the $k$ top directions, so to ensure that we have enough light blocks we need $m \gg k\epsilon_6 m/\epsilon_1$. This can be achieved by letting $\epsilon_6$ be appropriately small. $\qquad\square$

## A.4 Putting Everything Together

We now finish the proof of the theorem. Let $i \in \{1, \ldots, k\}$. We only need to show the existence in the row-space of $SA$ of a rank-$k$ space that forms a good low rank approximation of $A$ (up to an additive error of $\epsilon \|A\|_F^2$), since by Lemma A.2, SCW is guaranteed to compute an approximation at least as good.

Let $i \in \{1, \ldots, k\}$. If $\|A^L v_i\| \geq \epsilon_3 \sigma_i^2$, then by Lemma A.11, the row-space of $SA$ contains a direction $f_i$ such that $f_i^T v_i \geq 1 - \epsilon_1$. By including $f_i$ as one of the $k$ direction in our low rank approximation, we incur error at most $\epsilon \cdot \sigma_i^2$ from $\sigma_i^2$. The total error incurred on all such $v_i$'s together is at thus most $\epsilon_1 \|A\|_F^2$.

Next consider $v_i$ which has mass at least $\epsilon_3$ on the heavy rows, meaning $\|A^H v_i\|_2^2 \geq \epsilon_3 \sigma_i^2$. The total mass of $v_i$ on those heavy rows on which there is less than $\epsilon_7 \cdot \sigma_i^2$ mass is at most $n_H \epsilon_7 \cdot \sigma_i^2$, where $n_H$ is the number of heavy rows. By Lemma A.4 this is at most $\epsilon_7 \cdot \epsilon_5 m \cdot \sigma_i^2 \leq \epsilon_7 \cdot \epsilon_5 m \cdot \|A\|_F^2$, which is at most $\epsilon \|A\|_F^2$ for an appropriate choice of $\epsilon_7$, so we may ignore these heavy rows. For the remaining heavy rows, by Lemma A.10 we recover a $1 - \epsilon_9$ fraction of the mass of $v_i$ on them, thus incurring at most $\epsilon_9 \|A\|_F^2$ total additional error.

Finally we need to handle medium rows. Recall we use two partitions into blocks, one with $m'$ as above, and one with $m'' = m'/\epsilon_{10}$, where $\epsilon_{10} > \log(n)/(\epsilon_2 \epsilon_4^2 \epsilon_5)$. Since every medium row $a_i$ in the first partition does not satisfy eq. (2), it satisfies

$$\frac{\|a\|_2^2}{\|A\|_F^2} > \frac{\epsilon_2 \epsilon_4^2}{m' \log n},$$

and therefore by choice of $\epsilon_{10}$, it satisfies

$$\frac{\|a\|_2^2}{\|A\|_F^2} > \frac{1}{m'' \epsilon_5},$$

rendering the row heavy in the latter row classification. This implies no row is medium is both block partitions: if it is non-light with respect to $m'$, it is necessarily heavy with respect to $m''$. Since we recover a $(1 - \epsilon)$-fraction of the mass of each top-$k$ direction from light and heavy rows, and every row is either heavy or light in one of the two block partitions, we include a $1 - \epsilon$ approximation of each top-$k$ direction in the row-space of $SA$, and those approximate directions yield a low rank approximation of $A$ inside that row-space, with additive error at most $O(\epsilon) \cdot \|A\|_F^2$. Since SCW is guaranteed to find a solution at least as good (Lemma A.2), the proof is complete.

# B  Proof of Theorem 2.2

The theorem is implicit in the original works that developed the SCW algorithm (in particular, the correctness of the SCW algorithm was proved by designing a distribution over $S$ that with high probability satisfies conditions equivalent to the assumption of Theorem 2.2). Let us recap the proof for completeness, mostly following the presentation from [7].

By Lemma A.2, in order to prove the theorem, it suffices to exhibit any $B \in \mathbb{R}^{n \times m}$ of rank $k$ such that

$$\|A - BSA\|_F^2 \leq (1 + O(\epsilon)) \cdot \|A - A_k\|_F^2. \tag{6}$$

The lemma then guarantees that SCW returns an output at least as good as $BSA$.

The $B$ we choose is the following:

$$B = U_k (U_k^\top S^\top S U_k)^{-1} U_k^\top S^\top.$$

Our choice requires $U_k^\top S^\top S U_k$ to be invertible; we will later show this indeed follows from the conditions of the theorem. For brevity we will denote $Z = (U_k^\top S^\top S U_k)^{-1}$, rendering $B = U_k Z U_k^\top S^\top$.

In what follows we let $r$ denote the rank of $A$. We now prove eq. (6). We use the Pythagorean identity and decompose the left-hand side $\|A - BSA\|_F^2$ into the sum of projections onto and against the

top-$k$ column-space of $A$. The projection matrix onto is $U_k U_k^\top$ and the projection matrix against is $U_{r-k} U_{r-k}^\top$, thus:

$$\|BSA - A\|_F^2 = \|U_k Z U_k^\top S^\top SA - A\|_F^2$$
$$= \|U_k U_k^\top (U_k Z U_k^\top S^\top SA - A)\|_F^2 + \|U_{r-k} U_{r-k}^\top (U_k Z U_k^\top S^\top SA - A)\|_F^2$$
$$= \|U_k Z U_k^\top S^\top SA - A_k\|_F^2 + \|A - A_k\|_F^2.$$

Therefore in order to prove eq. (6) it suffices to prove:

$$\|U_k Z U_k^\top S^\top SA - A_k\|_F^2 \leq O(\epsilon) \cdot \|A - A_k\|_F^2. \tag{7}$$

Recall that the SVD of $A$ is $A = U\Sigma V^\top$, and let us decompose it into its top-$k$ and bottom-$(r-k)$ part as $A = U_k \Sigma_k V_k^\top + U_{r-k} \Sigma_{r-k} V_{r-k}^\top$. Now we have:

$$\|U_k Z U_k^\top S^\top SA - A_k\|_F^2 = \|U_k Z U_k^\top S^\top SA_k + U_k Z U_k^\top S^\top S(A - A_k) - A_k\|_F^2$$
$$= \|U_k Z U_k^\top S^\top S(A - A_k)\|_F^2$$
$$\leq \|U_k\|_2^2 \cdot \|Z\|_2^2 \cdot \|U_k^\top S^\top S(A - A_k)\|_F^2$$
$$= \|Z\|_2^2 \cdot \|U_k^\top S^\top S(A - A_k)\|_F^2,$$

where:

- The first equality is just by writing $A = A_k + (A - A_k)$.
- The second equality is by observing that since $Z = (U_k^\top S^\top SU_k)^{-1}$,
$$U_k Z U_k^\top S^\top SA_k = U_k Z U_k^\top S^\top SU_k \Sigma_k V_k^\top = U_k \Sigma_k V_k^\top = A_k.$$
- The third inequality is since $\|XY\|_F \leq \|X\|_2 \cdot \|Y\|_F$ for any matrices $X, Y$.
- The fourth equality is since $U_k$ has orthonormal columns and thus $\|U_k\|_2 = 1$.

Now we use the premise of the theorem, which we recall is

$$\|U_k^\top S^\top SU - I_0\|_F^2 \leq \epsilon.$$

By the column-wise additivity of the squared Frobenius norm, this implies both of the following:

1. $\|U_k^\top S^\top SU_k - I\|_F^2 \leq \epsilon \leq 1/9$. This implies $\|U_k^\top S^\top SU_k - I\|_2 \leq 1/3$, which in turn implies that all eigenvalues of $U_k^\top S^\top SU_k$ are between $2/3$ and $4/3$. (Note this implies that $U_k^\top S^\top SU_k$ is invertible, as promised earlier for our choice of $B$.) Equivalently, all eigenvalues of $Z = (U_k^\top S^\top SU_k)^{-1}$ are between $3/4$ and $3/2$. Thus, $\|Z\|_2^2 \leq 9/4$.

2. $\|U_k^\top S^\top SU_{r-k}\|_F^2 \leq \epsilon$. This yields,
$$\|U_k^\top S^\top S(A - A_k)\|_F^2 = \|U_k^\top S^\top SU_{r-k}\Sigma_{r-k}V_{r-k}^\top\|_F^2$$
$$\leq \|U_k^\top S^\top SU_{r-k}\|_F^2 \cdot \|\Sigma_{r-k}V_{r-k}^\top\|_2^2$$
$$\leq \epsilon \cdot \|A - A_k\|_2^2$$
$$\leq \epsilon \cdot \|A - A_k\|_F^2,$$
where we have observed that $\|\Sigma_{r-k}V_{r-k}^\top\|_2$ is the $(k+1)$th singular value of $A$, also written as $\|A - A_k\|_2$.

Plugging the bounds $\|Z\|_2^2 \leq 9/4$ and $\|U_k^\top S^\top S(A - A_k)\|_F^2 \leq \epsilon \cdot \|A - A_k\|_F^2$ above yields

$$\|U_k Z U_k^\top S^\top SA - A_k\|_F^2 \leq \tfrac{9}{4}\epsilon \cdot \|A - A_k\|_F^2,$$

proving eq. (7) and thus eq. (6), finishing the proof of the theorem.

## C  Additional Experimental Results

**Sparsity.** We evaluate the performance of SCW, 1Shot2Vec and FewShotSGD when the sparsity of the sketching matrix is increased. In particular, we evaluate each method when the number of non-zero entries in each column of the sketching matrix is 1 (as in the main text), and when it is 8. In both cases, the locations of the non-zero entries are chosen uniformly at random in each column, independently across columns. The results are give in Table 1.

Table 1: Error attained with sparsities 1 and 8 (denoted by the suffixes "–s1" and "–s8" respectively).

| Dataset | SCW–s1 | SCW–s8 | 1Shot2Vec–s1 | 1Shot2Vec–s8 | FewShotSGD–s1 | FewShotSGD–s8 |
|---------|--------|--------|--------------|--------------|---------------|---------------|
| Eagle   | 2.2    | 1.94   | 1.09         | 1.06         | 0.64          | 0.54          |
| Logo    | 0.47   | 0.42   | 0.21         | 0.26         | 0.13          | 0.1           |

Table 2: Evaluation of learned LRA in the transfer learning and mixed learning settings.

| Category | SCW  | FewShotSGD | Transfer | Mixed |
|----------|------|------------|----------|-------|
| Panda    | 8.39 | 4.34       | 5.21     | 5.04  |
| Okapi    | 7.22 | 5.14       | 5.26     | 5.41  |

**Transfer and mixed learning.** We evaluate FewShotSGD in the transfer and mixed learning settings. Suppose we are given two datasets $D_1$ and $D_2$. In the transfer learning setting, we learn the sketching matrix from the training set of $D_2$, and evaluate it on the test set of $D_1$. In the mixed learning setting, we learn the sketching matrix on the combined training sets of $D_1$ and $D_2$, and evaluate it on the test set of $D_1$.

The results given in Table 2 are evaluated on the Panda and Okapi categories of the Caltech-101 image dataset [4]. They show that on one of the categories (Okapi), transfer and mixed learning perform about as well as direct learning (i.e., learning on the training set of $D_1$ and evaluating on the test set of $D_1$), while on the other class (Panda), transfer and mixed learning perform better than the oblivious baseline SCW, but not as well as direct learning. We have also run these experiments on the Eagle and Logo datasets. The results (not included here) showed that on Logo, transfer and mixed learning performed no better than the oblivious baseline SCW, while on Eagle, transfer and mixed learning performed somewhat better than SCW, but not as well as direct learning (similar to the results for Okapi). The results reported here use the Caltech-101 dataset instead of Eagle and Logo, based on the premise that images from two semantically related categories of the same dataset might be more amenable to transfer and mixed learning than two unrelated videos.