# OpenReview forum: "Few-Shot Data-Driven Algorithms for Low Rank Approximation"
_NeurIPS.cc/2021/Conference — NeurIPS 2021 Poster_

### Official Review · Reviewer_sJ5B · 2021-07-08

**Rating:** 7
**Confidence:** 4

**Summary:**

Problem: Sketching based techniques (such as randomized algorithms, etc) is the go to approach for deriving a low rank approximation. Despite their power in terms of space and speed to deduce the solution, these algorithms induce large approximation errors, which regresses at the downstream tasks and need to be addressed
Solution: A data-driven approach for low rank approximation -- A low rank representation is learned by optimizing a frobenius loss function over the input training and approximation. There are some challenges with this approach. The proposed algorithm is less data hungry with a two step analytical approach.
Results: Significant gains in terms of accurate low rank approximations with less resource usage.


**Ethical Concerns:**

No ethical concerns to the best of my evaluation.

**Limitations And Societal Impact:**

Providing a low rank approximation is a handy technique, especially in the big data world. Given that, the proposed model clearly positions w.r.t the state-of-the-art methods such as IVY and addresses some of the limitations of the earlier method.

**Main Review:**

Distinction from state-of-the-art:
Providing a low rank approximation is a handy technique, especially in the big data world. Given that, the proposed model clearly positions w.r.t the state-of-the-art methods such as IVY and addresses some of the limitations of the earlier method.

Pros:
The paper is well written with a few to no grammatical errors, easy to follow and understand.

An effective analytical approximation which performs better even with small amounts of data.

Cons:
First of all, the abusive use of the word “latter” is really confusing, sometimes IVY is latter and other times it is the new method, please be consistent.

Since the proposed method is so fast, I wonder, what is the effect of multiple random initializations? If you repeat the experiment for multiple runs will the results still hold on an average? Even an average of these runs can help close the gap further, can you check?

I agree the method looks better over the state-of-the-art, however, can we add more value by exploring a different dimension in this space of research? That is, can the proposed approach be considered as a general enough for multiple inputs? Of course, the ranks are not the same for all the inputs. For example, what if you can synthesize the same rank benchmarks from different distributions and apply this method either individually or by mixing them? In the individual case, can the result of one benchmark be used as the initial point for a different benchmark (a form of transfer learning)? In the mixing case, can the solution be general enough?

Based on the response, willing to up the score.


### Post author response

Satisfied with the author response and increase the score to 7.

**Time Spent Reviewing:**

3 -- 4 hours

---

> ### Author Response · Authors · 2021-08-10
> **Response**
>
> We thank the reviewer for the feedback and comments.
>
> > the abusive use of the word “latter” is really confusing [...]
>
> We apologize for this lack of clarity, and we will revise the text accordingly.
>
> > Since the proposed method is so fast, I wonder, what is the effect of multiple random initializations? [...] In the individual case, can the result of one benchmark be used as the initial point for a different benchmark (a form of transfer learning)? In the mixing case, can the solution be general enough?
>
> Thank you for these interesting suggestions. We have run preliminary experiments with multiple random initializations. It seems that averaging multiple runs does not improve the accuracy, but that choosing the best of them using a validation set does lead to a noticeable improvement. Of course, the validation step also increases the running time and the sample complexity, and the trade-off with the improved accuracy needs to be studied more closely. We will look into this.
>
> As suggested by the reviewer, we have also run preliminary experiments in the transfer learning and mixed learning settings. Here there seems to be a difference between the datasets: Logo does not benefit from transfer learning (it performs no better than the oblivious baseline), while Eagle sees some improvement (though not as much as learning on its own training set). Similarly, in the mixed learning setting, the loss on Eagle improves compared to the oblivious baselines, but the loss on Logo does not.
>
> Related to the reviewer’s suggestion, transfer learning in the context of low-rank approximation has also been recently considered in Raab & Schleif, “Low-Rank Subspace Override for Unsupervised Domain Adaptation”, https://arxiv.org/pdf/1907.01343.pdf (we will include the reference). We have also run some experiments on the Caltech101 image dataset which they use. Here are, for example, the results for the two classes Panda and Okapi, each partitioned into a training set and a test set. Below we list the test loss for each one of the two classes, achieved by the following methods:
> (1) Oblivious LRA
> (2) Our algorithm, using learning on the training set of the same class
> (3) Transfer - Our algorithm, using learning on the training set of the other class
> (4) Mixed - Our algorithm using the combined training sets of both classes
>
> The results: (Oblivious / Ours / Transfer / Mixed)
>
> Panda: 8.39 / 4.34 / 5.21 / 5.04
>
> Okapi: 7.22 / 5.14 / 5.26 / 5.41
>
> As with the Logo and Eagle datasets, it seems that for one of the classes (Okapi) transfer and mixed learning work well, while for the other class (Panda) they work better than oblivious but not as well as learning on its own training set. This direction generally requires more exploration.
>
> In general we agree that there are many algorithmic avenues worth exploring here, and indeed, one of our motivations in this work (and a point we have attempted to argue in the text) is that the data-driven LRA problem, as formalized in [32], is an inherently interesting setting and opens the door for many algorithmic ideas beyond direct minimization of the empirical loss as done in [32].

---

### Official Review · Reviewer_oFt2 · 2021-07-14

**Rating:** 7
**Confidence:** 5

**Summary:**

This work is about the optimization of sparse sketching matrix in randomized low-rank matrix approximation. The data-driven low rank approximation algorithms are developed with better computational efficiency in the training phase. The experiments show the proposed algorithms are empirically advantageous over previous work with improved training time.

**Limitations And Societal Impact:**

Addressed

**Main Review:**

1.The sparse sketching matrix considered in this work contains a single non-zero entry in each column. This is not a widely-used sketching strategy and would lead to relatively large error in the produced randomized LRA. Allowing 8 nonzero entries in each column is more practical which makes similar accuracy of approximation to that employing Gaussian random matrix, as suggested in J. A. Tropp and Martinsson’s recent work.
The above sparse pattern (8 nonzeros in each column) in the sketching matrix should be considered in experiments. And, it seems the proposed method does not work in this scenario, while the method in [32] is able to handle it.

2. In Section 2.1, the idea of preserving as much of the Frobenius mass in 1Shot1Vec is not explained. By the way, do you assume that 2-norm of s_i is 1? Otherwise, it cannot infer that s_i should be the top left singular vector. This strategy and that randomly selecting singular vectors in 1Shot2Vec should be supported with more solid theoretic analysis and discussion.

3. In the experiments, what’s the Error in Fig. 1? Is that the formula after line 292 on Page 7? The value of that formula depends on the norm of the data matrix and is not very good. A dimensionless value to measure the error is preferable.
Although some matrix data are tested in Section 3, the application scenarios of them are not explained. And, the size of test matrix is small. Why is computing the LRA of them important?

Some minor concerns.
1. Algorithm descriptions should be given in Section 2.
2. Eq. (2) is wrong?
3. Step 2 of Alg. 1, alternatively, QR factorization can be applied.
4. The references are not complete, and several recent work very relevant to randomized LRA are missed, e.g.
[1] Randomized numerical linear algebra: Foundations and algorithms
PG Martinsson, JA Tropp - Acta Numerica, 2020 - cambridge.org
[2] Fast randomized PCA for sparse data
X Feng, Y Xie, M Song, W Yu… - Asian conference on …, 2018 - proceedings.mlr.press

------
After reading the authors' response and the paper again, I'm willing to raise my review score. This work is good and can be accepted, providing that the presentation is improved to address the review concerns.

**Time Spent Reviewing:**

4 hours

---

> ### Author Response · Authors · 2021-08-10
> **Response**
>
> We thank the reviewer for the feedback and comments.
>
> > In Section 2.1 [...] This strategy and that randomly selecting singular vectors in 1Shot2Vec should be supported with more solid theoretic analysis and discussion
>
> The theoretical analysis justifying these strategies is encompassed in Theorem 2.1 (the full proof is given in the appendix), and we have attempted to provide intuition in Section 2.3. To recap, choosing the top singular vector of a block ensures that the sketch accounts for “heavy” rows in the matrix (i.e., the ones with large l_2 norm relative to the total Frobenius norm), since such rows can be shown to be strongly aligned with the top singular vector of the block they are hashed to (Lemma A.10), while choosing a random singular vector of a block with the specified distribution ensures that the sketch captures the “light” rows (Lemma A.11), by relying on the notion of projection cost preserving sketches (PCPs, discussed starting in line 261). To this end, we first observe that the light portion of each block forms a PCP for the light portion of the input matrix with high probability (Corollary A.8), and then show that sampling singular vectors of that block according to the specified distribution (i.e., proportionally to their squared singular values) makes the sketch account for the input top directions which are aligned with light rows (this is Lemma A.11). The precise reason for this choice of distribution is that it preserves in expectation the projection length of every given direction in the block (cf. the equation after line 149 in the appendix). We will highlight more clearly the connection between the steps in Section 2.1 to their formal grounding in the analysis.
>
>
> > The sparse sketching matrix considered in this work contains a single non-zero entry in each column [...] Allowing 8 nonzero entries in each column is more practical [...] it seems the proposed method does not work in this scenario
>
> We emphasize that our algorithms can work with any number of nonzeros per column, not just one, and their analyses (Theorems 2.1 and 2.2) continue to hold. Generally, correctness is preserved since intuitively increasing the number of nonzero entries in the sketching matrix improves the sketch quality, and the running time is also preserved as long as the sketch remains sufficiently sparse, in particular if the number of nonzeros per column is O(1). We now discuss this in a bit more detail.
>
> In the few-shot SGD algorithm, one can apply gradient updates to any subset of entries in the sketching matrix, similarly to IVY in [32]. The algorithm remains the same as described in Section 2.2, and the correctness of Theorem 2.2 is preserved, since its proof (Section B in the supplementary material) does not hinge on the sparsity of the sketching matrix.
>
> In the 1-shot closed-form algorithms, there are two natural ways to increase the number of nonzeros per column. One is to concatenate several copies of a fixed sparsity pattern (we note that our algorithm 1Shot2Vec already does this and uses 2 nonzeros per column). Then, increasing the number of nonzeros amounts to just sampling more (say 8) singular vectors from each block, and the correctness of Theorem 2.1 is clearly preserved.
>
> The other way to increase the sparsity pattern is to choose a fixed constant number of entries (say 8) in each column to be nonzeros uniformly at random, independently across different columns. This amounts to hashing each row of the input matrix into 8 of the m blocks (where m is the sketching dimension) instead of just one block. For the proof of Theorem 2.1 to go through, we need (I) each block to contain at most one heavy row (where heavy and light rows are described in lines 250-252 as well as above, and defined in the supplementary material), (II) each heavy row to have strong signal compared to the “noise” created by the light (and medium) rows in its block (line 115 in the supplementary material). Both of these conditions continue to hold when each row is hashed into a constant number of blocks instead of one, as long as the sketching dimension is increased by an appropriate constant, and so Theorem 2.1 continues to hold.
>
> At the reviewer’s suggestion, we have run some experiments with sparsity 8. The preliminary results, listed below, appear similar whether the sparsity is 1 or 8, and do not seem to lead to a qualitative difference, even though more elaborate experiments may be in order. We will include them in the revised paper. In the results listed below, the algorithms are “Obv” - oblivious LRA, Our1Shot - the algorithm from Section 2.1, OurSGD - the algorithm from Section 2.2. The prefix “-s#” denotes the number of nonzero entries (sparsity) per column in the sketching matrix.
>
> Algorithm: Obv-s1 / Obv-s8 / Our1Shot-s1 / Our1Shot-s8 / OurSGD-s1 / OurSGD-s8
>
> Loss on Eagle: 2.2 / 1.94 / 1.09 / 1.06 / 0.64 / 0.54
>
> Loss on Logo: 0.47 / 0.42 / 0.21 / 0.26 / 0.13 / 0.1
>
>
> > By the way, do you assume that 2-norm of s_i is 1?
>
> Yes, according to the original SCW analysis (lemma A.2 in our appendix), its output depends only on the row-span of SA, so it can be assumed w.l.o.g. that the rows of S have unit norms. We will clarify this in the text.
>
> > In the experiments, what’s the Error in Fig. 1? [...]  the application scenarios of them are not explained [...] size of test matrix is small
>
> For the sake of compatibility with the literature, we use the same experimental setting introduced in [32] (and also used in [37]), including their choice of dataset, matrix dimensions, normalization, and error measurement.
>
> In particular, [32] normalized all matrices to have the same top singular value, which we reproduce. The notion of error, also inherited from [32], is the one after line 292, though we can also measure a different notion of error if one is proposed. The dimension of the matrices, combined with the number of matrices in the dataset, does seem to pose a computational bottleneck, as we report in lines 343--345, where sketch-based LRA gives an average speed-up by a multiplicative factor of 11.6.
>
> > Algorithm descriptions should be given in Section 2.
>
> Thank you, we will add them to the manuscript.
>
> > Eq. (2) is wrong?
>
> This is a typo, the training loss is defined on A_train rather than A_test, as in the right-hand side. Thank you for pointing this out.
>
> > Step 2 of Alg. 1, alternatively, QR factorization can be applied.
>
> Correct. For concreteness we have presented the algorithm as it was given in [16] and [32], but we will mention that other variants of it could also be used.
>
> > The references are not complete [...]
>
> Thank you, we will add the missing references.

---

### Official Review · Reviewer_hcCP · 2021-07-22

**Rating:** 6
**Confidence:** 4

**Summary:**

The paper proposes two simple methods to learn the sketching matrix in low-rank matrix approximation. They serve as strong baselines to compare against other data-driven methods that learn the sketching matrix from training data.
The first method (one-shot), 1Shot1Vec / 1Shot2Vec, sets the sketching matrix using the singular vectors of blocks of the matrices in the training data.
The second method (few-shot), uses a different objective from the usual data-driven low-rank approximation that avoids costly calls to the SVD.
The paper empirically validates these two approaches, showing that they are fast and are competitive with more complex methods.

**Limitations And Societal Impact:**

Yes

**Main Review:**

Strengths:
1. Both of the proposed methods are simple and fast. They avoid costly calls to the SVD. Moreover, the sketching matrix learned is interpretable, unlike the black-box nature of existing methods.
2. The modified objective of the few-shot method sheds light on the nature of low-rank approximation, pointing out more precisely what we need in a sketching matrix.

Weaknesses:
1. The motivation for the methods are not very strong. The paper motivates the method by arguing the existing methods requires auto-differentiation, GPUs, and long-training time. The first two conditions are now quite accessible in machine learning (or almost ubiquitous), so it's not clear to me when one would use the proposed methods instead of the existing methods.
2. The advantage compared to existing methods isn't very noticeable. The two proposed methods can be seen as better initialization for the sketching matrix before running SGD. However, as seen in Figure 1, after some iterations of SGD (around 3x the amount of time of the proposed methods), existing methods also find sketching matrices that are just as good. So it seems one would need to run SGD anyway, and I'm not sure how much benefits these few-shot algorithms would bring.

============== Post-rebuttal

Thank you for the response to my questions. That has helped me understand the method better.

I've increased my rating.

**Time Spent Reviewing:**

3

---

> ### Author Response · Authors · 2021-08-10
> **Response**
>
> We thank the reviewer for the feedback and comments.
>
> Regarding our runtime improvement, we believe that a 3x speed-up, without loss in accuracy, is a very significant improvement.
>
> Regarding GPUs: First, we note that all of our experiments are run on a GPU (including the non-SGD algorithms), so the reported speed-ups hold even if restricted to that setting. Nonetheless, indeed, we have also argued that their compatibility with CPU is an advantage. While GPUs are increasingly more accessible, especially to expert ML practitioners and large tech organizations, they are not yet as common as CPUs -- for example, while CPUs are built-in in any home computer, GPUs at present need to be purchased or comissioned separately, and are often managed as a scarce centralized resource even in large organizations.
>
> In a similar vein, as of now, auto-gradient functionality is still limited to a small handful of software packages and programming languages. We generally believe that eliminating the reliance of algorithms on specific software and hardware requirements is an important step toward scalability and wide-availablility, and consider it to be an important research direction.

---

### Author Response · Authors · 2021-08-24
**Review and discussion follow-up**

Dear reviewers,

Thank you again for your reviews and feedback on our work.

Reviewer oFt2: thank you for the updated review and score. We will make sure to address the points raised by the reviewers in the revised version of the paper.

Reviewers hcCP and sJ5B: we would appreciate it if you could please let us know if we have addressed your reviews satisfactorily, or whether there are additional outstanding questions or concerns, so that we have a chance to respond before the end of the discussion period on Sept 2.

---

### Decision · Program_Chairs · 2021-09-27

**Decision:**

Accept (Poster)

**Comment:**

The paper proposes a new data-driven approach for low rank approximation. Similar to previous works, it is based on the optimization of sparse sketching matrices, which leads to important reductions in computation and memory at test time. The proposed algorithm, significantly improves the computational efficiency in the training phase.

The authors provided a high quality rebuttal including new experiments and clarifying several points raised by the reviewers. All reviewers increased their scores by one and consequently all reviewers recommend accepting the paper.

All reviewers consider that the work makes an interesting contribution to the field. The proposed algorithm is simpler than previous work, but is able to achieve similar performance while being computationally more efficient. The method is well motivated from an analytic perspective providing several insights on the problem. For instance, the sketching matrix learned is interpretable and can be set with meaningful values extracted from the dataset.

Overall the AC considers that the paper makes a solid contribution to the field and recommends its acceptance. The authors should incorporate the several clarifications provided to the reviewers (in particular those of Reviewer oFt2).